# A Comprehensive Insight into Fungal Enzymes: Structure, Classification, and Their Role in Mankind’s Challenges

**DOI:** 10.3390/jof8010023

**Published:** 2021-12-28

**Authors:** Hamada El-Gendi, Ahmed K. Saleh, Raied Badierah, Elrashdy M. Redwan, Yousra A. El-Maradny, Esmail M. El-Fakharany

**Affiliations:** 1Bioprocess Development Department, Genetic Engineering and Biotechnology Research Institute, City of Scientific Research and Technological Applications (SRTA-City), Universities and Research Institutes Zone, New Borg El-Arab, Alexandria 21934, Egypt; elgendi1981@yahoo.com; 2Cellulose and Paper Department, National Research Centre, El-Tahrir St., Dokki, Giza 12622, Egypt; asrk_saleh@yahoo.com; 3Biological Science Department, Faculty of Science, King Abdulaziz University, P.O. Box 80203, Jeddah 21589, Saudi Arabia; rbadierah@kau.edu.sa (R.B.); rredwan@gmail.com (E.M.R.); 4Medical Laboratory, King Abdulaziz University Hospital, King Abdulaziz University, P.O. Box 80203, Jeddah 21589, Saudi Arabia; 5Protein Research Department, Genetic Engineering and Biotechnology Research Institute, City of Scientific Research and Technological Applications (SRTA-City), New Borg EL-Arab, Alexandria 21934, Egypt; dr.yousraadel@yahoo.com

**Keywords:** fungi, enzymes, structure and classification, function, applications

## Abstract

Enzymes have played a crucial role in mankind’s challenges to use different types of biological systems for a diversity of applications. They are proteins that break down and convert complicated compounds to produce simple products. Fungal enzymes are compatible, efficient, and proper products for many uses in medicinal requests, industrial processing, bioremediation purposes, and agricultural applications. Fungal enzymes have appropriate stability to give manufactured products suitable shelf life, affordable cost, and approved demands. Fungal enzymes have been used from ancient times to today in many industries, including baking, brewing, cheese making, antibiotics production, and commodities manufacturing, such as linen and leather. Furthermore, they also are used in other fields such as paper production, detergent, the textile industry, and in drinks and food technology in products manufacturing ranging from tea and coffee to fruit juice and wine. Recently, fungi have been used for the production of more than 50% of the needed enzymes. Fungi can produce different types of enzymes extracellularly, which gives a great chance for producing in large amounts with low cost and easy viability in purified forms using simple purification methods. In the present review, a comprehensive trial has been advanced to elaborate on the different types and structures of fungal enzymes as well as the current status of the uses of fungal enzymes in various applications.

## 1. Introduction

Enzymes play an important role in lowering the energy of activation and accelerating numerous biological reactions that are crucial in nourishing life without causing any permanent modifications. Enzymes are widely distributed in all forms of living organisms, including animal, plant, and microbial sources. The inability of plant and animal sources to satisfy industrial demands of enzymes directed attention to microbial sources [1]. Microbial enzymes production is faster, cost-effective, scalable, and amenable to genetic manipulations [2]. Among microbial sources, fungi represent an interesting source for industrial enzymes, sharing with more than one has of enzymes in the market [3]. Fungal enzymes are characterized by high production potency, easier purification and separation requirements, especially for filamentous fungi, and efficient catalysis with desired stability against harsh conditions [3]. Furthermore, the uses of fungi in a great number of traditional preparations, such as brewing and baking from ancient periods, offer a safe and firm context for their recent applications [3]. Enzymes have an important role in mankind’s challenges to use biological systems for a wide range of dedications. Currently, fungal enzymes are accounted for more than 50% of the total enzymes market [3]. This huge market share is largely attributed to a few species of *Aspergillus*, *Trichoderma*, *Rhizopus*, and *Penicillium* genera that fulfill the commercial-scale requirements for enzymes production (Table 1). The fast-growing enzymes market forced the continuous search for novel enzymes producers with desirable industrial characteristics. Recently, mushroom cultivation has represented a promising competitor in enzymes production in terms of higher productivity and lower invested cost [4]. Attributed to the nutritional value and health benefits of the edible mushroom, its global production pace is very fast, recording 8.99 million tons in 2018 [4]. Mushrooms are mostly saprophytic fungi and derive their nutrient from surrounding cellulosic material; hence, lignocellulolytic enzymes are highly characterized in most mushroom cells, including cellulases, β-glucosidases lignin peroxidases, and laccases [5,6]. This diverse enzymatic machine enhanced the ability of mushrooms to grow in diverse low-cost agriculture wastes inspiring their industrial cultivation for nutrition and enzyme production [4].

In general, enzymes are classified under seven classes, including transferases, oxidoreductases, lyases, hydrolases, ligases, isomerases, and translocases [49]. Fungal cells revealed both intracellular and extracellular enzymatic activity. Fungi are considered natural decomposers, and consequently, they are allowed to produce a great number of extracellular enzymes needed for bioconversion of a wide range of substrates and complexes [50]. Other extracellular enzymes may participate in fungal protection against hazardous compounds that exist naturally or result from the substrate hydrolysis process. Commercially important fungal enzymes belong to hydrolases and oxidoreductases. Among extracellular fungal enzymes, laccases and peroxidases, such as manganese peroxidase and lignin peroxidase, represent the two main subclasses of fungal enzymes that have been investigated for the degradation of xenobiotics and removal of harmful phenolic components from industrial environment and wastewater [51]. The current review discusses the different types of enzymes produced by fungi involved in many biotechnological applications. The structure of the fungal enzymes is also discussed. Numerous biotechnological uses of fungal enzymes, including industrial, medical, bioremediation, and environmental applications, are critically reviewed.

## 2. Fungal Enzymes Structure and Function

Enzymes are proteins in nature, formed originally from a long chain of amino acids linked together through peptide linkage. Once the primary amino acid chain is formed, subsequent multistep processes take place, collectively known as post-translation modification, where this long chain is modified into its three-dimensional structure (Figure 1). Sequence and nature of the amino acid in various enzymes are varied and have an obvious relationship to enzyme activity and stability. Not all enzyme molecules get involved in the catalytic reaction, but only very few amino acids of (2–4) are called the active site [52], where the remaining part (called Apoenzyme) does not participate directly. Most enzymes are synthesized inactive and activated to holoenzyme (the functional form of an enzyme) with different organic/or inorganic factors. The organic enzyme activators are usually non-proteinaceous molecules known as coenzymes, e.g., vitamins, whereas the inorganic molecules are called cofactors, e.g., zinc, iron, calcium, and copper [52]. The accumulation of enzymes end products may terminate the enzyme activity in a process called feedback inhibition. The feedback enzyme inhibition is a regulatory mechanism initiated by cells to control the amount needed from the enzyme end products [53]. Generally, enzymes are very specific catalysts; however, some enzymes have broad substrate specificity and could work upon multiple substrates. Different theories were proposed to elucidate the enzyme–substrate interaction and recognition, but the most popular one is the lock and key theory proposed by Emil Fischer [54]. This theory elucidates the enzyme and substrate recognition as unique specific complementation between enzyme active site and substrate resembling lock and key. Unfortunately, this theory could not explain the transition state stabilization, where enzyme configuration was altered to fit the proceeding catalytic process [52,55]. To explain these configuration changes in the enzyme active site, a flexible enzyme active site concept was proposed by Koshland, establishing the enzymes Induced Fit Model. In this model, the active site is flexible, where the substrate can modify its orientation to achieve the perfect fit [56]. In Michaelis–Menten’s theory, two phases of catalytic reaction were proposed, where in the first phase, an enzyme-substrate complex was established, and then, the enzyme and products were liberated in the second phase [57].

## 3. Fungal Enzyme Nomenclature and Classifications

Nowadays, fungal enzymes represent more than one-half of the market enzymes sharing in the different industrial sectors [3]. Due to the vast number of enzymes discovered to date, many criteria were proposed to organize and classify enzymes, including their origin, structure, or substrate. The first rational enzymes nomenclature and classification system were firstly proposed in the 1950s, based upon the enzyme catalytic reaction [58]. The International Union of Pure and Applied Chemistry (IUPAC) adopted the new system, where enzymes are classified into six groups: oxidoreductases, transferases, hydrolases, lyases, isomerases, ligases, and translocases [59]. This classification is a numerically based system where each catalytic reaction (enzymes group) is given a number called commission number (EC number) to specify its enzyme members [52].

### 3.1. Oxidoreductases (EC 1)

Oxidoreductases are enzymes catalyzing the oxidation/reduction (redox) reaction through the transfer of an electron from donor to an electron acceptor. Generally, redox reactions are numerous in the living cells and included in many vital biological processes such as tricarboxylic acid cycle, glycolysis, amino acid metabolism, and oxidative phosphorylation reactions; hence, oxidoreductases are among the housekeeping enzymes essential for cell’s life and activity. Recently, fungal oxidoreductases were characterized to play a major role in pathogenicity and protection against host defense mechanisms [60]. Yu et al. reported the importance of oxidoreductase-like protein Olp1 in *Cryptococcus neoformans* for sexual reproduction through enhancing the meiotic division and its essential role in the pathogenicity by protecting the fungal cells from lithium–ion toxicity [60]. Benzoquinone oxidoreductase is another reported fungal virulence factor produced by *Beauveria bassiana* (Entomopathogenic fungus) to overcome the exposure to benzoquinone from the host cell [61]. To date, more than twenty classes of oxidoreductases have been recognized; however, the most studied and reported oxidoreductases include dehydrogenases (transfer of hydrogen to an electron acceptor), oxygenase (the final electron acceptor is oxygen), and peroxidases (the final electron acceptor is peroxides) [2]. Commercially applied oxygenases include monooxygenases, dioxygenases, and laccases. Fungal laccases are copper-containing oxygenases and play an important role in fungal physiological growth, including sporulation, pigmentation, and aid in plant pathogenesis [62]. Laccases have a high potential to oxidize a wide range of phenolic and non-phenolic compounds, independently of any cofactors [2,62]. The high efficiency of fungal laccases, in addition to cofactor-independence nature, nominated them to several industrial applications such as stain removal [63], paper industry [64], medical applications [65], and biosensor manufacturing [66]. Though microbial laccases were reported from bacteria [67,68], white-rot fungi are the main source of laccases with commercial importance [69]. Peroxidases are oxidoreductases where peroxide compound (H_2_O_2_) acts as the final electron acceptor [70]. The majority of peroxidases recognized to date are metal-dependent enzymes, especially iron; however, metal-independent peroxidases were also reported [70]. Two of the most extensively reported peroxidases are lignin peroxidases (LiP) and manganese-peroxidases (MnP). Basidiomycetes, especially white-rot fungi, are characterized by their high ability to degrade lignin attributed to the extracellular production of lignin peroxidases and manganese-peroxidases [71]. Lignin peroxidases (LiP) catalyze the oxidation of a wide range of phenolic compounds; hence, they have several industrial applications [72], in contrast to manganese-peroxidase, which has narrow substrate specificity and low oxidation potential [70].

### 3.2. Transferases (EC 2)

Transferases catalyze the transfer or exchange of certain groups (amino group) among many compounds [73]. This process has a vital role in creating essential amino acids for protein synthesis in all cells [74]. Some transferases such as glutathione transferase may assist in environmental and oxidative stress tolerance in pathogenic fungi [75]. Glutathione transferases catalyze the reversible transfer (incorporation/elimination) of glutathione group to/from different compounds [76]. Incorporation or elimination of glutathione group resulted in more soluble fewer toxic compounds that can be handled by the fungal cells. Fungal glutathione transferases secreted by fungal cells during host pathogenesis to tolerate oxidative stress created by host and/or toxic wood-derived molecules [77,78]. Fungal glutathione transferases have wide substrate specificity; therefore, they have the potential to detoxify many toxic xenobiotic compounds [76,78,79]. The production of glutathione transferases was reported from many fungi, including *Alternaria brassicicola* [78], *Phanerochaete chrysosporium* [80], and *Saccharomyces cerevisiae* [81]. Another commercially important transferase is fructosyltransferase, which catalyzes the transformation of sucrose into fructooligosaccharides [82]. Fructooligosaccharides are commercially important due to their numerous health benefits as lowering blood pressure and flourishing the growth of beneficial microflora while inhibiting the other pathogenic ones [83]. The sweetness effect and viscous texture of fructooligosaccharides encourage their application in the food industry [82]. Fructosyltransferase production was widely reported from the Aspergillus genus, including *Aspergillus oryzae* [84], *Aspergillus niger* [85], *Aspergillus aculeatus* [86]. However, [82] reported applying *Aureobasidium pullulans* for fructosyltransferase commercial production.

### 3.3. Hydrolases (EC 3)

Hydrolases are the most extensively studied groups of enzymes; they catalyze the hydrolysis of their substrate through the addition of water. To date, hydrolases represent the most commercially marketed enzymes due to their wide application in different industrial sectors [3,52,87]. Fungal proteases, amylases, lipases, and cellulases represent the most commercially demanded enzymes [88]; hence, continuous improvement in production efficiency with lowered cost are mandatory [89].

Proteases play an important role in fungal physiology to digest extracellular large peptides and also in defense mechanisms against attaching pathogens [90]. Based upon the amino acid in the enzyme active site, proteases could be categorized into different types, including serine, asparagine, cysteine, aspartic, and metalloproteases [90]. Serine and metalloprotease are the most studied types among all proteases and are usually produced from microbial origins [90]. Fungal proteases are privileged over that of bacteria for easier purification process and less hazardous application [91,92]. Filamentous fungi, especially that of *Aspergillus* sp., are characterized by their high capacity for protease production [89,93,94]. Other fungal genera also reported for their potency regarding proteases production, including *Penicillium* sp. [95], *Fusarium* sp. [96,97], and *Pichia farinosa* [98].

Amylase is the world premiere in enzyme production for commercial application and was firstly applied medicinally in treating digestive disorders [99]. Amylases could be classified into α, β, and γ-Amylases depending on the attaching site in the starch molecules and the nature of the resulting products. α-Amylases are calcium-dependent metalloenzymes that act randomly on the starchy substrates yielding maltose and maltotriose from amylose or glucose and dextrin from amylopectin [87]. β-Amylases hydrolyze 1,4-glycosidic bonds in the carbohydrate chain, yielding one maltose unit at a time. They are extensively important in plants, especially in the seed ripping process, but they are also reported from the microbial origin [100,101]. γ-Amylases resemble the other two types of amylases in hydrolysis activity toward 1,4-glycosidic linkages, unlike the two forms characterized with 1,6-glycosidic linkages hydrolysis activity and preferring acidic environment pH 3 [87]. *Aspergillus niger* is considered the potent commercial α-Amylase producer among all filamentous fungi [87]. Many other fungi were reported for their capacity to produce different types of amylases, including *Aspergillus oryzae* [102], *Aspergillus terreus* [103], *Fusarium solani* [104], and *Penicillium citrinum* [105].

Lipases are a group of hydrolytic enzymes that act by hydrolysis of triacylglycerol yielding fatty acid and glycerol [106]. Lipases also catalyze the reverse reaction by esterification of glycerol and fatty acid. Microbial lipase is produced by many bacteria *Bacillus licheniformis* [107], *Geobacillus thermodenitrificans* [108], *Pseudomonas aeuriginosa* [109], and fungal cells including *Aspergillus niger* [110,111], *Penicillium verrucosum* [112], *Fusarium solani* [113], *Arthrographis curvata*, and *Rhodosporidium babjevae* [114]. The remarkable specificity and thermal stability, in addition to the easier recovery procedure of the extracellular fungal lipase over other microbial sources, attracted great attention from the application point of view [115,116,117]. Lipases are implemented in vast commercial applications, including detergents and cosmetics additives, fine chemical production, medical application, paper pitching, leather de-fating, wastewater treatment, and biodiesel production [118,119,120]. The application of lipase in biodiesel production, as an ecofriendly alternative for traditional fuel, intensifies the research in diminishing the production cost and enhancing the enzyme efficiency. In this regard, solid-state fermentation of agricultural wastes represents a step forward cost reduction [110], while protein engineering has been recently applied for enhancing lipase efficiency [121].

The continual necessity for renewable ecofriendly fuel sources intensified the research in cellulose-degrading enzymes. The worldwide abundance of cellulosic material represents a promising ideal source for energy, hence the extensive studies and application of cellulolytic enzymes [48], ranking them among the most worldwide marketing enzyme [122]. Cellulose, hemicellulose, and lignin are the main components of most agricultural wastes representing 40–50%, 25–30%, and 15–20% for cellulose, hemicellulose, and lignin, respectively [48]. Most fungi have the complete enzymatic system (Endoglucanases, Cellobiohydrolases, B-glucosidases, and Xylanases) to degrade this complex cellulosic material for nutrition [123]. *Trichoderma reesei* is widely applied for the commercial production of cellulases [48], but other fungi also represent potent cellulase producers, including *Aspergillus niger* [124], *Saccharomyces cerevisiae* [125], and *Aspergillus brasiliensis* [126]. Xylan, a complex polysaccharide, is also a major component of hemicellulose; hence, xylanases play an important role in the efficient hydrolysis of plant cellulolytic material [127]. Regarding the diverse and complex structure of Xylan, its hydrolysis required a group of synergistically working enzymes (xylanolytic system) for complete degradation [128]. Filamentous fungi are characterized by the required xylanolytic system for complete xylan degradation, especially that of *Trichoderma reesei* [128,129], *Aspergillus oryzae* [130], and *Aspergillus flavus* [131]. Vast numbers of other hydrolytic enzymes are produced by fungi with wide commercial applications, including fungal phytase and pectinase. Phytase act upon phytic acid (plant inorganic phosphorus), dissolving its content of insoluble inorganic phosphorus makes it available for different physiological processes. Fungal phytase is available for different commercial applications from *Aspergillus niger* [132]. Pectinase also has important industrial applications, especially in food and paper industries, produced commercially through fungi [133].

The worldwide availability of chitin, representing the second abundant natural polymer next to cellulose, renders chitinases among the most important enzymes [134]. Fungal chitinases belong to hydrolytic enzymes with an extracellular role in chitin decomposition, where intracellularly involved in cell wall lysis and reconstruction in addition to protein deglycosylation [135]. Chitin is one of the main structures in pests’ cell walls and some plant-pathogenic fungi [136]. Hence, chitinases are widely applied in agriculture for plant infection control to overcome the conventional chemical fungi/insecticidal hazardous [137]. In addition, the chitin degradation products (chitosan and chitooligosaccharides) are implemented for various medical applications [134]. Trichoderma viride chitinases were reported with acidoplic, thermostable, and wide chitinolytic activities for commercial applications under the name of Usukizyme [134,138]. Chitinases were reported within various fungal species, including thermostable chitinases (maximum activity between 60 and 55 °C) through *Myceliophthora thermophila* [139], and *Humicola grisea* [140] with thermotolerant/mesophilic chitinases (maximum activity between 40 and 37 °C) through *Penicillium oxalicum* [137] and *Trichoderma koningiopsis* [141].

### 3.4. Lyases (EC 4)

Lyases are a group of enzymes that catalyze the addition or elimination reaction. The result of this type of addition or elimination is a new compound with either cyclic structure or new double bonds [73]. Though lyases usually require only one substrate to act in one direction, two substrates are necessary to reverse this reaction. Normally, lyases have very limited substrate specificity; however, the independence of expensive cofactors encourages their commercial applications in many industries. One of the most reported lyases is pectin lyase catalyzes random degradation of pectin through the elimination of water molecules yielding one unite of galacturonan with unsaturated double bonds [142]. Pectin lyase plays a major role in fungal pathogenicity, though degrading the pectic materials holding the plant cell wall together [142]. The role of pectin lyase in fungal pathogenicity was reported in many plant pathogenic fungi, such as *Clonostachys rosea* [143], *Cylindrocarpon destructans* [142], and *Penicillium canescens* [144]. Alginate lyase represents another important fungal lyase; it catalyzes the degradation of alginate to defined oligosaccharides. Alginate lyase was reported in fungi, including terrestrial fungus *Aspergillus oryzae* [145] and marine fungus *Paradendryphiella salina* [146].

### 3.5. Isomerases (EC 5)

Isomerases are a group of enzymes that catalyze the rearrangement of its substrate structure through the interchange of a specific group within the same compound [52]. One of the most widely studied and applied fungal isomerases is glucose/xylose isomerase, representing the third marketing enzyme with protease and amylase [147]. Glucose/xylose isomerase reversely converts D-glucose and D-Xylose into D-fructose and D-Xylulose, respectively [148,149]. Though glucose/xylose isomerase was extensively reported from many bacteria as *Bacillus licheniformis* [147], *Serratia marcescens* [150], *Caldicellulosiruptor bescii* [151], and *Streptomyces lividans* [152], the fungal enzyme was reported from a few species of *Aspergillus* genus [153]. Marshall and Kooi reported for the first time the ability of xylose isomerase from *Pseudomonas hydrophila* to convert D-glucose into D-fructose [154]. Production of high fructose corn syrup through glucose isomerase has wide application in the food industry [148]. With emerging of white biotechnology and biofuel concepts, great attention was directed toward xylose isomerase as a tool to overcome the xylose accumulation issue. The hydrolysis of lignocelluloses is a prerequisite for the growth of *Saccharomyces cerevisiae* and biofuel production. The monosaccharide in hydrolysate liquor is a mixture of glucose (60–70%) and xylose (30–40%) [155], while *Saccharomyces cerevisiae* does not have the necessary system to assimilate the xylose [84]. Taking into account that assimilation of xylose in bacteria is one-step process and independent in any cofactor, contrary to that of fungi which is multi-steps and dependent on expensive cofactors [155], many trials were conducted to insert the gene for bacterial xylose isomerase in the genome of *Saccharomyces cerevisiae* and were first achieved with the xylose isomerase gene from *Termus thermophilus* [156]. Recently, the expression of xylose isomerase from *Clostridium phytofermentans* in *Saccharomyces cerevisiae* greatly enhanced xylose fermentation [157].

### 3.6. Ligases (EC 6)

Ligases, in general, are those classes of enzymes that catalyze the joining of two compounds through the formation of new bonds. Different types of ligases may be classified according to the nature of bonds established in the new compounds, such as carbon–carbon bonds, carbon–nitrogen bonds, carbon–sulfur bonds, carbon–oxygen bonds, nitrogen–metal bonds, and phosphoric–ester bonds [73]. To date, most of the reported ligases are intracellular, where their main function is modifying the cellular nucleic acid content. The unique structure and mechanism of some fungal ligases represent a promising target for developing new antifungal drugs. The tRNA ligase (Trl1) represents a good candidate for such drugs. The tRNA ligase (Trl1) plays an important role in repairing the RNA breaks in fungal cells [158]. Fungal tRNA ligases (Trl1) were reported to have a similar structure in different pathogenic fungi [159,160]. Most bacteria and mammalian cells have different biochemical mechanism and structure of tRNA ligase from that reported in fungi [160]; therefore, an extensive research directed toward fungal tRNA ligases (Trl1) as a target for new antifungal drugs [160]. Apart from the molecular level, ligases were also reported to have other roles in different fungi. Ubiquitin ligases (E3s) catalyze the ligation of ubiquitin (low molecular weight protein) to the target protein (a process known as ubiquitination). Conjugation of protein to ubiquitin determines its fate inside the cell, whether to be degraded, activated, or relocated [161,162]. Unicellular fungi, including *Saccharomyces cerevisiae*, represent an ideal model for fully understanding the evolutionary background for the ubiquitination system [163].

## 4. Application of Fungal Enzymes

The present review discusses the advancement in fungal enzyme technology for different applications. A comprehensive list of fields, fungal source of enzymes, enzymes names, and the wide range of their application is conceded. Table 1 and Figure 2 give an overview of applications of fungal enzymes in different application fields.

### 4.1. Industrial Applications

#### 4.1.1. Food and Beverage

Food enzymes are also provided by microbial fermentation (bacteria, fungi, yeasts, actinomycetes, and algae), for which fungal strains are widely used. The application of fungal enzymes in food production offers a promising approach to enhance the preservation and shelf life of foods without affecting the characteristics of the organoleptic and nutritional content of foods [164]. The common fungal enzyme used in food applications is l-asparaginase which is considered GRAS (generally recognized as safe), which is used as a food additive to prevent the formation of acrylamide generated from the reaction between the food components as amino acids and reducing sugars at high temperatures and low humidity. This response often creates desirable color, flavor, and aroma compounds [165,166]. To minimize acrylamide levels in food, l-asparaginase assists with mitigation strategies from two aspects, intervention with the Maillard reaction or by converting l-asparagine into l-aspartic acid (non-toxic) to prevent precursors formation without altering the nutritional value, quality, or taste of the final product [167,168,169]. Single or consortium fungal enzymes were applied to improve the quality, clarity, and yield of fruit juice that ensure consumer appeal [11].

#### 4.1.2. Pulp and Paper

The government’s policies impose sustained pressure on the paper industry to protect the environment from hazards and diseases by replacing the chemical bleaching processes with greener ecofriendly alternatives. Enzymes such as xylanases and laccases obtained from fungi have the ability for bio-bleaching of agriculture wastes-based pulps by cleaving the β-1, 4 backbone of the complex plant cell wall for papermaking [170,171]. Ecofriendly papermaking depends on the two processing, Bio-pulping followed by bio-bleaching. Bio-pulping is the pre-treatment of agriculture-waste-based pulp by lignin-degrading enzymes before the routine pulping process. The process of bio-pulping is technologically feasible and cost-effective, reduces energy costs, and improves paper strength properties [172]. Filamentous fungi, especially members of white-rot fungi are the main source for lignin-degrading enzymes, whoever those with cellulases-free activity are important for the paper industry [173]. *Ceriporiopsis subvermispora*, *Ganoderma australe Phellinus pini*, and *Phlebia tremellosa* are three white-rot fungi with potent cellulases-free lignin-degrading activities suitable for the bio-pulping process where *Phanerochaete chrysosporium* is the model white-rot fungi with a complete enzymes system for lignin mineralization [174]. Bio-bleaching refers to the complete removal of lignin from bio-pulping wastes by the fungal enzyme to achieve white and bright pulp appropriate for papermaking. Bio-bleaching refers to the complete removal of lignin from bio-pulping wastes by the fungal enzyme to achieve white and bright pulp appropriate for papermaking. In addition to improving paper quality, bio-bleaching reduces the applied chemicals that diminish effluent toxicity and pollution [170]. Xylanases from *Fusarium equiseti* MF-3 and *Talaromyces thermophilus* were successfully applied for the bio-pulping process with a significant reduction in the organo-chloro compounds applied in the conventional bleaching process [17,175].

#### 4.1.3. Textile

Fungal enzymes play an important role in enhancing textile processing as well as valorizing the waste generated from streams of fibers, textile, and clothing manufacturing processes to value products [176]. Of the 7000 known enzymes, only about 75 are commonly used in textile manufacturing [177]. Xylanases are among the most common enzymes used for their effectiveness in textile processing at several stages, including desizing, bio-scouring (removing waxy material from fibers), suitable for coloration with low conductive fabrics, bio-bleaching, and bio-finishing of cellulosic fabrics to improve the textile quality [22]. The main mechanism for using fungal enzymes in textile processing, removal or degradation of undesirable components such as lignin, pectin, starch, excess H_2_O_2_ after bleaching, waxes by xylanases, or polygalacturonases or amylases form a fibers-based textile. Cellulases for cotton remove projecting fibers selectively, and this action is known as bio-polishing. In contrast, the textile wastes contain 35–40% of cellulosic fibers, which is utilized as a carbon sole for the production of bioethanol, biocellulose, etc., after being degraded by cellulases enzyme originated from *Trichoderma reesei* ATCC 24,449 [20]. Another application of fungal enzymes, which degrade the textile dyes by using *Pleurotus ostreatus* and *Pleurotus eryngii*, had the activity of laccase, lignin peroxidase, manganese peroxidase, xylanase, cellulase, and lipase for removal of days after 24 h [178]. The fungal enzymes applications textile is illustrated in Figure 3.

### 4.2. Environmental Applications

#### Bioremediation

With the continuous increase in the human being population, production, and consumption, industrial and agricultural wastes emerged as cumulative pressing environmental challenges that threaten human health and welfare. Waste management through fungal enzymes arises as promising solutions toward green and sustainable environments with the potential for generating new valuable products in a less consuming energy approach [179,180]. Currently, polyethylene-based materials (plastic packaging) are heavily consumed commercially attributed to their relatively low cost, and hence environmentally accumulated at a very high rate. Polyethylene-derived material revealed a high resistance for biological bioremediation; therefore, developing an effective approach for polyethylene removal is a pressing necessity from an environmental perspective. In this regard, a mixture of fungal oxidoreductases including laccase, manganese-peroxidase, lignin peroxidase, and unspecified peroxygenase was firstly studied (in silico) for eco-friendly polyethylene degradation [181]. The molecular docking results indicate the importance and essential role of laccase and unspecified peroxygenase in the polyethylene degradation process [181].

In the same regard, the widespread contamination with polyaromatic hydrocarbons from industrial effluents represents a huge challenge for human and animal health. In nature, polyaromatic hydrocarbons usually react to surrounding compounds to form exceptional multi-complex structures, such as aliphatic alkanes/alkenes, or halogenated hydrocarbons, potentially more toxic than the original structures [182]. Long-term exposure to polyaromatic hydrocarbons associated with many health complications included hepatic and renal toxicity and lung function damage, with a high rate of cancer development [183]. To overcome these problems, *Aspergillus oryzae* and *Mucor irregularis* were isolated from crude oil and reported as hydrocarbon degradation by secretion of several enzymes such as laccase, manganese peroxidase, and lignin peroxidase [184].

Recently, fungal laccases were widely applied for the bioremediation of synthetic textile dyes [185] by altering the dye molecules into non-colored, safer, and ecofriendly structures. Dyes-based industries (textile, paper, paint, and tannery) usually discharge their wastewater into water systems without any prior treatment and are heavily loaded with many unused synthetic dyes. These dyes imply many serious impacts in the aquatic environment, including increasing water turbidity and inhibiting the growth of some organisms like algae. In addition, most of these dyes are carcinogenic with high hepatocellular and renal toxicity upon long-term exposure [186]. There are two major mechanisms for decolorization of dyes using fungi, which are biosorption and biodegradation. The biosorption is essentially committed for dead cells. In this mechanism, the association of physiochemical interactions such as adsorption, deposition, and ion exchange is capable of decolorization. The biodegradation is particularly committed for living cells because they can produce the lignin-modifying enzymes, laccase, manganese peroxidase, and lignin peroxidase [187]. Fungal species (including *Alternaria*, *Aspergillus*, and *Cladosporium*) are capable of producing extracellular enzymes (lignin peroxidases, manganese peroxidases, and laccases) that can biodegrade complex synthetic dyes (Congo red, Poly R-478, Methyl green, indigo carmine, etc.) which widen the industrial applications of fungal enzymes toward the sustainable environment [35,37,188].

With vast developments in agriculture techniques, the worldwide accumulation of lignocellulosic agricultural wastes also represents a considerable environmental challenge. As discussed above, fungi naturally genetically inherited diverse enzymes (hydrolases and oxidoreductases) for the ecofriendly valorization of cellulosic material into valuable industrial products like bioethanol. Collectively, the application of fungal enzymes revealed dual sustainable benefits from an industrial quality and environmental perspective.

### 4.3. Biomedical Applications

#### 4.3.1. Antimicrobial

Fungi are well known for the production of antimicrobial agents, industrial enzymes, and microbial biomass. They have evolved the ability to synthesize novel metabolites, including natural products and enzymes, which could be used for novel antimicrobial. The antimicrobial activity of the fungal enzyme may be classified into two categories: direct and indirect antimicrobial agents. First, fungal enzymes such as cellulases, amylases, and lipases were used directly for antimicrobial activity [40], while indirect antimicrobial activity depended on generating intermediate compounds with antimicrobial activity through fungal enzymes [42]. Direct antimicrobial activity includes rupturing the cell membrane, resulting in the loss of intracellular elements necessary for life. Furthermore, fungal enzymes stop DNA synthesis, inhibit essential bacterial enzymes, block bacterial receptors, and inhibit the electron transport chain [189,190], as illustrated in Figure 4. In addition, many fungal enzymes have the ability to inhibit both DNA and RNA viruses. The purified oyster mushroom laccase extracted from *Pleurotus ostreatus* showed potent antiviral activity against hepatitis C Virus (HCV) [191]. Other fungal enzymes, such as ribonucleases, can exert an alternative antiviral activity against hepatitis B virus (HBV) [192] and human immunodeficiency virus type 1 (HIV-1) [193]. Regarding many fungal crude extracts and their enzymes, the antiviral effect of fungal lectin against HIV was the most investigated virus. El-Maradny et al. [194] indicated that the purified Trichogin protein from *Tricholoma giganteum* was found to inhibit the HIV reverse transcriptase (RT) enzyme with the lowest IC_50_ (half maximal inhibitory concentration) value determined to be 83 nM [195], followed by the purified lectin from *Russula delica* with IC_50_ value of 0.26 [75] and the purified laccase from *Tricholoma mongolicum* with IC_50_ value of 0.62 μM [196]. Lectins bind with viral envelope glycoproteins and inhibit the entry of many viruses, such as HIV, thus inhibiting virus attachment and blocking its entry via inducing the virus conformational rearrangements [197]. Ma et al. [198] revealed that the purified lectin from *Agrocybe aegerita* was found to be a nontoxic protein and served as an effective adjuvant with influenza vaccine against H9N2 viruses in infected mice model through binding with viral envelope glycoprotein.

#### 4.3.2. Anticancer

Fungi produce a diverse set of natural products as secondary metabolites, including polyketides, nonribosomal peptides, terpenes, chitosan-oligosaccharides, polyketide–terpene hybrids, and polyketide–nonribosomal peptide hybrids by the action of fungal enzymes. Those secondary metabolites have been a rich source for therapeutics used as antibiotics (penicillin), cholesterol-lowering agents (lovastatin), and immune suppressants (cyclosporine) [199]. Chitosan-oligosaccharides are low-molecular-weight chitosan derivatives with interesting anticancer applications, which are produced from the *Trichoderma harzianum* chitinolytic system by the action of extracellular enzymes. As anticancer compounds, chitosan-oligosaccharides inhibited the growth of cervical cancer Hela cells at mild concentration and reduced the survival cells rate significantly [42]. The specific mechanism of chitosan preventing tumor growth is unclear, although it might be linked to chitosan electrostatic charges, alterations in tumor cell permeability, and modulation of tumor factor expression, including metalloproteinase-9 or/and vascular endothelial growth factors [200]. Endophytic fungi anticancer toxal (Paclitaxel) production has also been reported from *Penicillium aurantiogriseum* (NRRL 62431) [201], *Cladosporium oxysporum* [202], *Phoma medicaginis* [203] by the action of fungal enzymes. Paclitaxel is the world’s first billion-dollar anticancer drug and is used to treat breast, ovarian, and lung cancer [204]. Taxol was the first microtubule targeting drug to be disclosed in the literature, and its major mode of action was to disrupt microtubule dynamics, resulting in mitotic arrest and cell death [205], as presented in Figure 5. Additionally, various fungal enzymes act as prodrug activators for anticancer drugs [206]. The enzyme α-L-rhamnosidase, derived from Acremonium species, is used in the one-step rhamnosylation of anticancer drugs such as 2′-deoxy-5-fluorouridine, cytosine arabinoside, and hydroxyurea [207]. Surprisingly, these prodrugs only had a lethal impact on breast cancer MDA-MB-231 cells after being exposed to -L-rhamnosidase, showing that they are delivered to tumor locations specifically. The yeast cytosine deaminase enzyme is another model of enzyme-activated prodrug treatment. This enzyme has been studied in the conversion of the non-toxic prodrug 5-fluorocytosine (5-FC) to the cytotoxic 5-fluorouracil (5-FU) [208].

#### 4.3.3. Antioxidant

Antioxidants are important compounds that enhance the living organisms to avoid oxidative stress originated by reactive oxygen species, which are harmful compounds, causing damage and altering the structure of the carbohydrates, nucleic acids, lipids, and proteins [209]. Fungi can profoundly participate in the production of antioxidants metabolites, including phenolics, flavonoids, β-glucans, and steroids [210]. The main mechanism of fungal enzyme antioxidants such as catalases, peroxidases, superoxide dismutases, and transferases becomes activated to defend the drastic condition by converting reactive oxygen species into oxygen and water [211]. Phenolic compounds are natural antioxidants and bind to the cellulose matrixes of the plant cell walls [212]. Fungi are the widely promising organisms that produce lignocellulolytic enzymes responsible for plant cell walls degradation [213].

## 5. Toward Safe, Sustainable Production and Application of Fungal Enzymes

Industrial implementation of the fungal enzymes represents a sustainable, eco-friendly, and energy-saving solution for many environmental and quality aspects compared to the currently applied conventional chemicals/physical approaches [214]. However, some biosafety aspects should be considered in the fungal enzyme applications and productions. Recently, a sensitization toward the fungal enzymes was estimated to be 1–15% among populations, while in the enzyme production workers, the risk increased [215,216]. Bakery and detergent-packaging employees are clear examples of the prevalence of fungal enzymes allergy [216]. From the commercial point of view, the application of genetically modified fungal cells for enzyme production is more reliable and applicable in terms of energy-saving and nutrient consumption [217]; however, the biosafety and regulations for genetically modified enzymes are still challenging. Many criteria govern the application of genetically modified organisms for enzyme production, including genetic stability, absence of any potential virulence, and antibiotic resistance factors [218].

The huge cell mass and byproducts generated during enzyme production through fungi also represent a considerable environmental risk and imply an additional cost for management and control [219]. Several attempts were applied to implement such byproducts as bio-fertilizers [220] or in synthetic dye and heavy metal removal [221]. Nikkilä and his colleagues evaluated the nutritional quality of the aqueous and alkaline treated fungal-biomass resulting as a byproduct from the enzyme industry. The extracted fungal-biomass samples revealed high contents of both polysaccharides and glycoproteins with carbohydrate and amino acid contents of 37.5 and 27.6%, respectively, for alkaline-treated fungal biomass, where the aqueous ex-traction yielded 6.6 and 61.3% of carbohydrate and amino acid contents, respectively [219]. In the same regard, mushroom cultivation at the industrial level produces huge amounts of a byproduct called spent mushroom substrate comprises a mixture of remaining substrate and mushroom mycelia [222]. This byproduct represents a potential challenge for the mushroom growing industry and to the environment. As per literature, there is growing interest in the implementation of the spent mushroom substrate in valuable bioproducts and enzymes synthesis [223]. The enzymes production capacity of six *Trichoderma* spp. were evaluated using 36% cellulases-hydrolyzed spent mushroom substrate. As a sole substrate, spent mushroom substrate supported several enzymes production through *Trichoderma atroviride* isolate T42, compared to *Aspergillus niger*, including β-glucosidases (five isoforms) and twelve isoforms of both endocellulases and xylanases [224].

## 6. Conclusions and Future Perspectives

In recent times, it has been recognized that the production of microbial enzymes is retaining an advanced topic of innovation. The use of enzymes is still better than the use of chemical processes as they can carry out several reaction types of importance with more considerable efficiency under mild circumstances. Among microbial enzymes, fungal enzymes are at the best locations, owing to their ability to be carried out in large quantities and much faster as well as at affordable costs. Production of the fungal enzyme has had an incredible influence not only on industrial sectors but also on biochemical synthesis processes and healthcare services. The progress of biotechnology in these fields offers different approaches to reduce the cost of enzyme production economically, enhance methods of gene discovery, and decrease the time of enzymes production through accelerating the growth of production hosts. Furthermore, the increased access toward production systems of the fungal enzyme also raises the availability and sustainability of manufacturing dealing with low-value and bulk products. Usage of microbial enzymes, especially fungal enzymes, has significantly established their importance in processing and quality improvement. Rapid progress has happened in enzyme marketing over recent periods, owing to the development of industrial and biotechnological uses. Present developments in protein modeling and engineering, besides advancements in recombination technology, have developed enzyme commercialization and production. Three main applications led to an increase in the marketing of fungal enzymes, including technical, food industry, and animal feed sectors. The technical sector includes enzymes that are used in search and analysis, such as cellulases, proteases, and amylases. Moreover, these enzymes are comprehensively utilized in the textile, leather, starch, detergent, and personal care, as well as paper and pulp industries. The second big market sector is for the use of fungal enzymes in the food industry, such as lipases and pectinases. The third essential one is the feed sector which includes several enzymes such as xylanases, β-glucanases, and phytase. There is no doubt that in the near future, the utilizing of fungal enzymes will be expanded rapidly in several fields such as food processing, bio-pulping, chemical conversions, carbohydrate conversions, cleaning, animal feed, and food additives, as well as used in detoxification and the removal of pollutants from the environment. In addition, in the case of the use of other processes other than enzymatic uses, various industries produce byproducts, including wastes and contaminants, which cause environmental problems and can be harmful to nature. However, the use of fungal enzymes, which break down some particular substrates into amino acids that are useful, biodegradable, and naturally can be easily recycled in the surrounding environment. Additionally, the use of enzymatic processes leads to increasing overall yields and the formation of well-made products with avoiding undesirable byproducts. All these reasons give fungal enzymes advanced importance for enabling many industrial areas to be increased efficiency. Moreover, the usage of fungal enzymes offers the development and production of more friendly main products in the environment and acting via using less raw materials, energy, and water. In addition, the usage of fungal enzymes in many processes leads to the production of less waste and hazardous products. Although there has been a great revolution in exploring novel fungal enzymes, it is also an important demand for assuming a new screening program of novel strains to optimize the enzyme production with new characteristics, as the current identified fungi not more than 5% of the fungal flora. The combination of the modem knowledge of biotechnology and molecular biology can offer a better-quality yield and an efficient approach with novel features.

## Figures and Tables

**Figure 1 jof-08-00023-f001:**
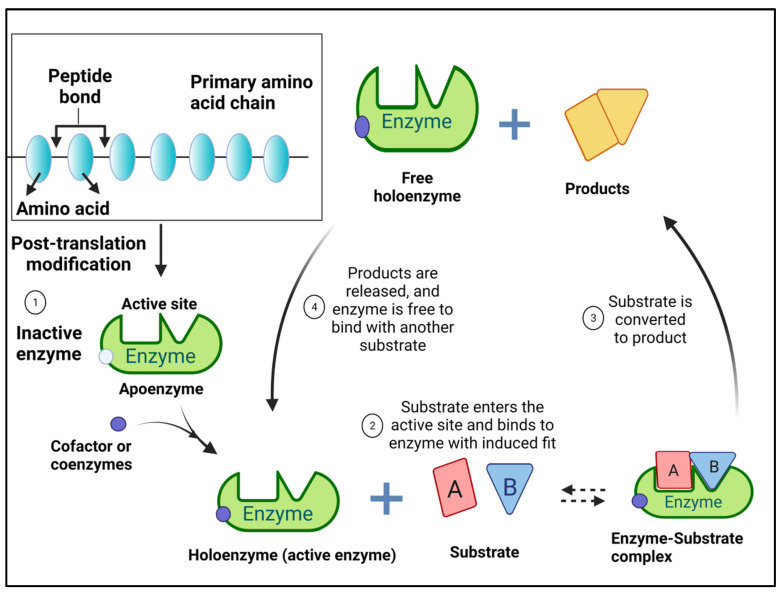
Schematic illustration for enzyme structure, activation, and steps of enzyme and substrate interaction.

**Figure 2 jof-08-00023-f002:**
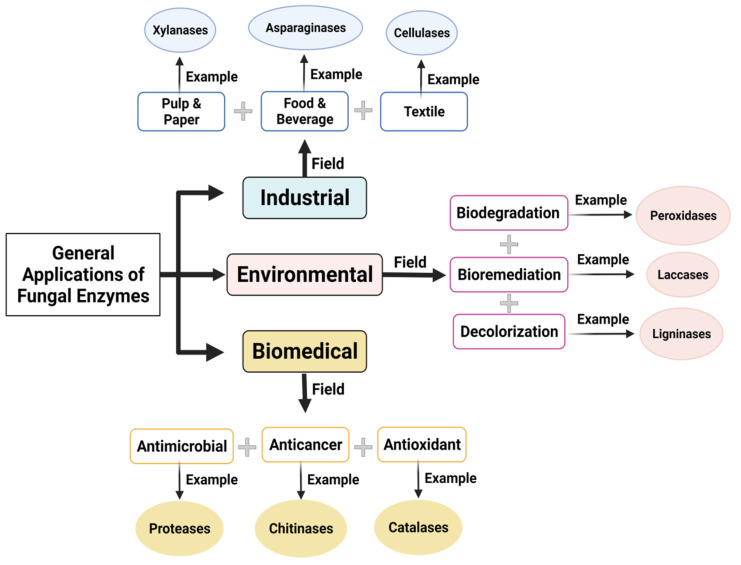
Schematic diagram of general application of fungal enzymes.

**Figure 3 jof-08-00023-f003:**
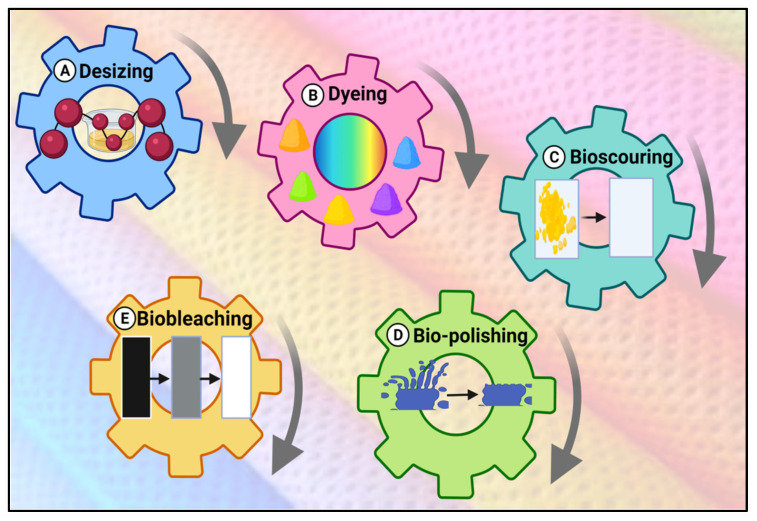
Fungal enzyme applications for textiles. (**A**) Desizing agents such as amylases; (**B**) dyeing and dye diffusers such as proteases; (**C**) textile processing and bioscouring of cotton fibers such as pectinases; (**D**) biofinishing and biopolishing such as cellulases; and (**E**) biobleaching and enzymatic rinse process after reactive dyeing such as peroxidases.

**Figure 4 jof-08-00023-f004:**
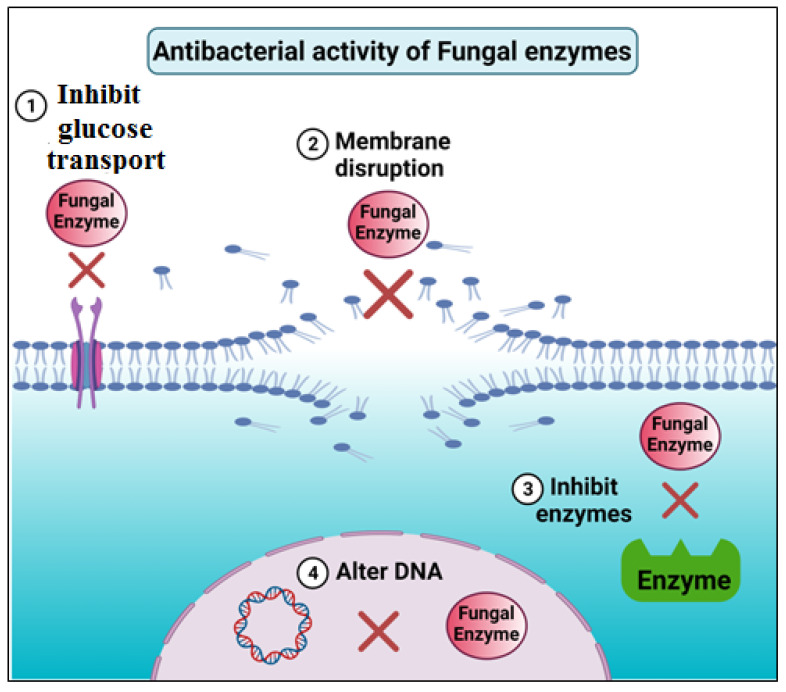
Schematic illustration represents the molecular mechanisms of the main antibacterial effects of fungal enzymes, including 1. alter membrane permeability and inhibition of glucose transport, 2. bacterial membrane damage and disruption, causing cytoplasm leakage, 3. inhibit bacterial enzymes, and 4. inhibit or alter cell division and the DNA replication process.

**Figure 5 jof-08-00023-f005:**
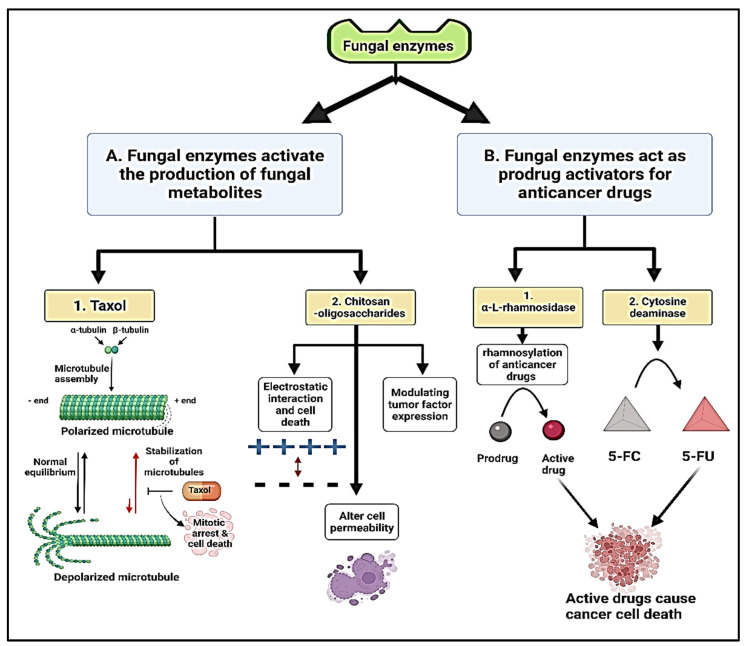
Schematic illustration represents the mechanism of anticancer effect of the fungal enzymes, 5-fluorocytosine (5-FC), and 5-fluorouracil (5-FU).

**Table 1 jof-08-00023-t001:** General application of fungal enzymes in different fields.

Application	Field	Fungal Name	Enzyme Name	Enzyme Use	Ref.
Industries	Food and beverage	*Aspergillus oryzae*, *Aspergillus oryzae* CCT 3940, and *Fusariumculmorum* ASP-87	l-asparaginases	Reduce the acrylamide formation in potato chips or French fries, bakery products, and coffee by degradation of l-asparagine	[7,8,9]
		* Myceliophthora thermophilia *	Laccases	Dough conditioner	[10]
		*Aspergillus niger* DFR-5	Xylanases	Improve yield and clarity of pineapple juice	[11]
		*Aspergillus niger*	Pectinases	Improve the quantity of the extracted Orange juice	[12]
		*Talaromyces leycettanus*	Pectinases	Efficiency in pectin degradation from grape juice	[13]
	Pulp and paper	*Pycnoporus cinnabarinus*	Laccases	Improve the brightness and strength properties of the pulp	[14]
		*Trametes villosa*	Laccases	Internal sizing of paper by use of laccase and hydrophobic compounds	[15]
		*Trichoderma reesei* QM9414	Xylanases	Eco-friendly of biobleaching of Kraft pulp of sugarcane straw	[16]
		*Trichoderma viride* VKF-3, *Fusariumequiseti* MF-3, and *Aspergillus japonicus* MF-1	Cellulases, xylanases, laccases, and lipases	Treatment enhances the brightness, deinking and reduces the heavy metals in the newspaper pulp	[17]
		*Rhizopus oryzae* MUCL 28168 and *Fusarium solani*	Tannases	Detoxification of coffee pulp by reduction of caffeine and tannins	[18]
		*Aspergillus niger*, *Phanerochaete chrysosporium*, and *Pycnoporus cinnabarinus*	Feruloyl esterases, Mn^2+^-oxidizing peroxidases, and laccases	Decrease the final lignin content of flax pulp and improvement of pulp brightness	[19]
	Textile	*Aspergillus niger* CKB and *Trichoderma reesei* ATCC 24449	Cellulases	Textile waste hydrolysis for recovery of glucose and polyester	[20,21]
		*Trichoderma longibrachiatum* KT693225	Xylanases	Desizing, bioscouring, and biofinishing of cellulosic fabrics (textile) without adding any additives	[22]
		* Aspergillus *	Amylases	Desizing of cotton fibers by removal of starch from the surface of textile fibers	[23]
		* Candida orthopsilosis *	Pectinases	Bioscouring of cotton fibers	[24]
		*Aspergillus niger* and *Penicillium*	Glucose oxidases and catalases	Removal of hydrogen peroxide from cotton bioprocessing	[25]
		*Chaetomium globosum* IMA1	Lignin peroxidases laccases and manganese peroxidases	Decolorization of the industrial textile effluent	[26]
Environment	Biodegradation	* Irpex lacteus * and *Pleurotusostreatus*	Manganese-peroxidases and laccases	Biodegradation of chlorhexidine and octenidine as antimicrobial compounds used in oral careproducts	[27]
		* Marasmius * sp.	Laccases	Degrade lignin by oxidizing the phenolic and non-phenolic compounds to produce dimers, oligomers, and polymers	[28]
		*Mucor circinelloides*	Lipases, laccases, and peroxidases	Biodegradation of diesel oil hydrocarbons	[29]
	Bioremediation	*Penicillium* sp.	Enzymatic reduction by the *mer* operon	The fungal enzyme could detoxify mercury (II) by extracellular sequestration via adsorption and precipitation	[30]
		*Coriolopsis gallica*	Laccases	Bioremediation of pollutants such as bisphenol, diclofenac, and 17-a-ethinylestradiol in real samples from the AQUIRIS wastewater	[31]
		*Aspergillus flavus* FS4 and *Aspergillus fumigates* FS6	Extracellular enzymes	Fungal consortium used for removal of chromium and cadmium	[32]
		*Pycnoporus sanguineus*	Laccases	Fugal laccase was immobilized on calcium and copper alginate/chitosan beads and used for the removal of 17 a-ethinylestradiol	[33]
		*Thermomyces lanuginosus*	Chitinases	Biocontrol agent against larvae of *Eldana saccharina* and fungi of *Aspergillus* sp., *Mucor* sp., and *Fusarium verticillioides*	[34]
	Decolorization	*Phanerochaete chrysosporium* CDBB 686	Ligninolytic enzymes	Decolorization of Congo red, Poly R-478, and Methyl green	[35]
		* Geotrichum candidum *	Peroxidases and laccases	Decolorization of methyl orange, Congo red, trypan blue, and Eriochrome black T	[36]
		*Coprinopsis cinerea*	Laccases	High indigo dye decolorization	[37]
		*Trametes* sp. SYBC-L4	Laccases	Decolorization of Congo red, aniline blue, and indigo carmine	[38]
		*Phanerochaete chrysosporium*	Manganese peroxidase	Decolorization of AO7 or CV pigment	[39]
Biomedical	Antimicrobial	32 different isolated fungi identified by morphological characteristics and internal transcribed spacer sequence analysis	Amylases, proteases, pectinases, xylanases, cellulases, chitinases, and lipases	Antimicrobial activity against pathogenic organisms by agar diffusion assays	[40]
		*Aspergillus oryzae* and *Aspergillus flavipes*	Proteases	Production of bioactive peptides from bovine and goat milk and the generated peptides tested against bacteria and fungi	[41]
		*Trichoderma harzianum*	Chitinases	Degradation of chitosan to form chitosan-oligosaccharides and used as antimicrobial against pathogenic organisms	[42]
	Anticancer	*Trichoderma viride* AUMC 13021	Chitinases	Antitumor efficiency of chitinase against different types of cancer cell line	[43]
		*Trichoderma harzianum*	Chitinases	Chitosan-oligosaccharides used as anticancer compounds, which inhibited the growth of cervical cancer cells at concentration of 4 mg/mL and significantly reduced the survival rate of the cells	[42]
		*Aspergillus terreus*	l-asparaginases	The synthesized zinc oxide conjugated l-asparaginase nanobiocomposite on MCF-7 cell line using MTT assay	[44]
	Antioxidant	*Aspergillus flavus*	Catalases	Antioxidant system plays a crucial role in fungal development, aflatoxins biosynthesis, and virulence	[45]
		*Pleurotus columbinus*, *P. foridanus*, *Aspergillus fumigatus*, and *Paecilomyces variotii*	Peroxidases and catalases	Production of enzymatic antioxidant from peels of banana, pomegranate, and orange	[46]
		*Chytridiomycetes* sp.	Ligninases	During biodegradation of lignin, the fungi synthesize bioactive compounds such as mycophenolic acid, dicerandrol C, phenyl acetates, anthrax quinones, benzo furans, and alkenyl phenols that have antioxidant activities	[47,48]

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
