# Peer review of "A Comprehensive Insight into Fungal Enzymes: Structure, Classification, and Their Role in Mankind’s Challenges"

_jof, 2021, doi:10.3390/jof8010023_

Round 1

Reviewer 1 Report

I would appreciate if you focused on the novelty of fungal enzymes.

Author Response

Comments and Suggestions for Authors

I would appreciate if you focused on the novelty of fungal enzymes.

Response

Firstly: we appreciate your reviewing feedbacks and your careful reading of our manuscript, and interest in our study. Secondly, we appreciate the opportunity to clarify our research novelty. We have revised all the manuscript and have made all necessary changes according to your recommendation.

Reviewer 2 Report

The paper is well written and the topi is interesting.

My general comments on the paper is that it is sometimes superficial in topics presentation. It can be improved.

Few concepts are redundant. Indeed. The fact that fungal enzymes represented more than one half of the market enzymes is repeated three times.

Enzymes names are sometimes singular or plural. I personally prefer for instance peroxidases than peroxidase, since many isoforms can be present in a single class. Please revise the text to be uniform.

Not all the topics are presented with the same depth. For instance in the EC1 chapter, authors did present some examples of application for laccases but not for peroxidases. On the other hand, the actual role of the enzymes for the fungus in not always presented. For the transferases, this part is briefly mentioned, but for oxidoreductases this should be implemented. They are important for lignin and cellulose degradation, but they have also other functions.

Line 368: please rephrase

Ch. 4.2 should be revise. Biodegradation, bioremediation and decolorization are not distinct concepts. They are all bioremediation processes, just applied against different matrices.

Please be aware that there are some minor typos, e.g. line 121, line 123, line 133 etc.

Author Response

Comments and Suggestions for Authors

The paper is well written and the topic is interesting.

My general comments on the paper is that it is sometimes superficial in topics presentation. It can be improved.

Response

All manuscript was revised, improved, and provided with more information and recent references. 

Few concepts are redundant. Indeed. The fact that fungal enzymes represented more than one-half of the market enzymes is repeated three times.

Response

Thank you for your comments.

The manuscript was revised, and the redundant concepts were edited or removed.  

Enzymes names are sometimes singular or plural. I personally prefer for instance peroxidases than peroxidase, since many isoforms can be present in a single class. Please revise the text to be uniform.

Response

The names of the enzymes were unified in all the manuscript in plural form.   

Not all the topics are presented with the same depth. For instance, in the EC1 chapter, authors did present some examples of application for laccases but not for peroxidases. On the other hand, the actual role of the enzymes for the fungus in not always presented. For the transferases, this part is briefly mentioned, but for oxidoreductases, this should be implemented. They are important for lignin and cellulose degradation, but they have also other functions.

Response

Thank you for your valuable comments.

All these comments are included the revised manuscript.

Line 368: please rephrase

Response

This line was rephrased.

Ch. 4.2 should be revise. Biodegradation, bioremediation and decolorization are not distinct concepts. They are all bioremediation processes, just applied against different matrices.

Response

Thank you for this consideration, the Ch. 4.2 was rewritten and developed with recent references.

Please be aware that there are some minor typos, e.g. line 121, line 123, line 133 etc.

Response

All errors were corrected.

Reviewer 3 Report

Manuscript "A Comprehensive Insight into Fungal Enzymes: Structure, Classification and Their Role in Mankind's Challenges" by
Hamada El-Gendi, Ahmed K. Saleh, Yousra A. El-Maradny and Esmail M. El-Fakharany provides an overview of the state of the art in biotechnology-related research into the production of enzymes derived from fungi.
The work presents a lot of data, however, for better presentation and usefulness, it requires some improvements. I hope the authors will find it useful to improve the quality of the illustrations, to supplement them with the necessary captions and explanations.
It is also necessary to more clearly characterize the necessity and importance of this work, complementing the missing information. The most important part of this work is the summarizing table, which in some cases contains links to reviews, but does not contain significant experimental articles.
I hope the corrections can be made relatively quickly and efficiently, as the work is interesting and can be useful.

I consider it necessary to expand the introduction by supplementing it with information and necessary references regarding the development and growth prospects of the mushroom enzyme market.
It is equally important to emphasize the importance of the environmental aspect of the application, in particular for sustainable development and ecology, which also requires references to relevant sources. The biosafety aspect of the production of enzymes and the resulting waste is not reflected in any way.

Despite the extensive review of the literature, it seems to me important to take into account also the materials below.

Lebreton, A., Bonnardel, F., Dai, Y. C., Imberty, A., Martin, F. M., & Lisacek, F. (2021). A Comprehensive Phylogenetic and Bioinformatics Survey of Lectins in the Fungal kingdom. Journal of Fungi7(6), 453.

Mathew, G. M., Madhavan, A., Arun, K. B., Sindhu, R., Binod, P., Singhania, R. R., ... & Pandey, A. (2021). Thermophilic Chitinases: Structural, Functional and Engineering Attributes for Industrial Applications. Applied Biochemistry and Biotechnology193(1), 142-164.

Santacruz-Juárez, E., Buendia-Corona, R. E., Ramírez, R. E., & Sánchez, C. (2021). Fungal enzymes for the degradation of polyethylene: Molecular docking simulation and biodegradation pathway proposal. Journal of Hazardous Materials411, 125118.

The schema presented in Figure 1. does not allow us to assess what is at stake, what processes are occurring where or not. There is no decoding of designations and symbols, in the upper right corner it is not clear what is happening with the product marked with an asterisk - if this is the main product, it should be obvious. The scheme needs to be decorated more accurately, and it is better to redraw it by adding specifics.

The schema shown in Figure 2. is clearer. However, the resolution of the images is insufficient, there are no descriptions of the meanings of colored frames or arrows, there is no explanation of what the arrows mean in each particular case, directed in one or two directions. Readers should not guess, in addition, if you provide data on various variants of enzyme activity, either an indication of what was meant or a reference is required. It is reasonable to designate such things, for example, by numbers, next to the arrow and decipher in the signature to the diagram.

The presentation in Figure 3. also requires a detailed caption and / or even a table, since this is a scientific review and not a popular science article.
Figure 4 also does not contain the proper information in the caption, since the figures are an independent scientific product and should be easily perceived even without reading the article. In addition, the abbreviation FE is extremely unfortunate, another abbreviation should be used, such as EF, which will not be misleading. It is also advisable to write the names of the processes indicated in the image, marking them with numbers or letters.

Figure 5 is at first glance more correct, but the signature is also uninformative, in addition, there is no obvious information in the figure about the stage at which the cell cycle stops, and this is important and for this there are appropriate designations (usually an outward arrow indicates an exit from the cycle, or an arrow directed, for example, to the G0 phase indicates the importance of influencing this phase).

The references in Table 1 should be expanded. It is advisable to use not only links to reviews, but also use the data of the original articles.
It is desirable to place the table before the conclusion. In addition, it requires discussion and conclusion. In this case, it is desirable to reflect the state of research in each area (relative to each enzyme). Consider problems in learning and / or production.

Author Response

Comments and Suggestions for Authors

Manuscript "A Comprehensive Insight into Fungal Enzymes: Structure, Classification and Their Role in Mankind's Challenges" by
Hamada El-Gendi, Ahmed K. Saleh, Yousra A. El-Maradny and Esmail M. El-Fakharany provides an overview of the state of the art in biotechnology-related research into the production of enzymes derived from fungi.

  • The work presents a lot of data, however, for better presentation and usefulness, it requires some improvements. I hope the authors will find it useful to improve the quality of the illustrations, to supplement them with the necessary captions and explanations

Response

Thank you for your comments.

All figures were improved.

It is also necessary to more clearly characterize the necessity and importance of this work, complementing the missing information.

Response

We appreciate your reviewing feedbacks and your careful reading of our manuscript, and interest in our study. We appreciate the opportunity to clarify our research novelty. We have revised all the manuscript and have made all necessary changes according to your comments and recommendations.

The most important part of this work is the summarizing table, which in some cases contains links to reviews, but does not contain significant experimental articles.

Response

All cited references in table 1 were revised and recited according to the original article.

I hope the corrections can be made relatively quickly and efficiently, as the work is interesting and can be useful.

Response

The manuscript was revised and improved.

I consider it necessary to expand the introduction by supplementing it with information and necessary references regarding the development and growth prospects of the mushroom enzyme market.

Response

Mushroom enzymes marketting, importance and types were included in the introduction. 

It is equally important to emphasize the importance of the environmental aspect of the application, in particular for sustainable development and ecology, which also requires references to relevant sources.

Response

The environmental aspects were added.

The biosafety aspect of the production of enzymes and the resulting waste is not reflected in any way.

Response

Thank you for the comments.

The enzyme biosafety, regulation and consideration were included in the manuscript under the title:  Toward safe and sustainable production and application of fungal enzymes.

 Despite the extensive review of the literature, it seems to me important to take into account also the materials below.

Lebreton, A., Bonnardel, F., Dai, Y. C., Imberty, A., Martin, F. M., &Lisacek, F. (2021). A Comprehensive Phylogenetic and Bioinformatics Survey of Lectins in the Fungal kingdom. Journal of Fungi7(6), 453.

Mathew, G. M., Madhavan, A., Arun, K. B., Sindhu, R., Binod, P., Singhania, R. R., ...& Pandey, A. (2021). Thermophilic Chitinases: Structural, Functional and Engineering Attributes for Industrial Applications. Applied Biochemistry and Biotechnology193(1), 142-164.

Santacruz-Juárez, E., Buendia-Corona, R. E., Ramírez, R. E., & Sánchez, C. (2021). Fungal enzymes for the degradation of polyethylene: Molecular docking simulation and biodegradation pathway proposal. Journal of Hazardous Materials411, 125118.

Response

Two of the proposed articles are really interesting and enriched the information of the manuscript, hence they were included, discussed in the current manuscript.

The schema presented in Figure 1. does not allow us to assess what is at stake, what processes are occurring where or not. There is no decoding of designations and symbols, in the upper right corner it is not clear what is happening with the product marked with an asterisk - if this is the main product, it should be obvious. The scheme needs to be decorated more accurately, and it is better to redraw it by adding specifics.

Response

Figure 1 was improved.

The schema shown in Figure 2. is clearer. However, the resolution of the images is insufficient, there are no descriptions of the meanings of colored frames or arrows, there is no explanation of what the arrows mean in each particular case, directed in one or two directions. Readers should not guess, in addition, if you provide data on various variants of enzyme activity, either an indication of what was meant or a reference is required. It is reasonable to designate such things, for example, by numbers, next to the arrow and decipher in the signature to the diagram.

Response

Figure 2 was improved.

The presentation in Figure 3. also requires a detailed caption and / or even a table, since this is a scientific review and not a popular science article.

Response

Figure 3 was improved.

Figure 4 also does not contain the proper information in the caption, since the figures are an independent scientific product and should be easily perceived even without reading the article. In addition, the abbreviation FE is extremely unfortunate, another abbreviation should be used, such as EF, which will not be misleading. It is also advisable to write the names of the processes indicated in the image, marking them with numbers or letters.

Response

The caption of Figure 4 was improved.

Figure 5 is at first glance more correct, but the signature is also uninformative, in addition, there is no obvious information in the figure about the stage at which the cell cycle stops, and this is important and for this there are appropriate designations (usually an outward arrow indicates an exit from the cycle, or an arrow directed, for example, to the G0 phase indicates the importance of influencing this phase).

Response

Figure 5 was improved.

The references in Table 1 should be expanded. It is advisable to use not only links to reviews, but also use the data of the original articles. (Ahmed saleh)

Response

All cited references in table 1 were revised and recited according to the original article

It is desirable to place the table before the conclusion. In addition, it requires discussion and conclusion. In this case, it is desirable to reflect the state of research in each area (relative to each enzyme). Consider problems in learning and / or production.

Response

Table 1 was placed before conclusion section.

Reviewer 4 Report

Your review approach an important and an actual, theme: enzymes from fungi. Despite this, the manuscript (Ms.) must be deepen in some sections (3 and 4), other eliminated (section 2) or edited (section 1). As yeasts are completely absent from the text (just one or two notes in table 4), the title should reflect that.

Point 1- change several sentences: remove or change some statements.

Point 3 – deepen the information – give examples, conditions, compare species, phyla, for instance; add chitinases.

Point 4 - deepen the various sub-sections; basidiomycetous fungi are completely absent from the text.

I did not understand the inclusion of 4.3.2 and 4.3.3 in the text. Of course, in point 4.3.2, enzymes are behind the synthesis of almost all substances – from that point of view all fungal enzymes are important.

The information in Table 4 is not presented/explained, introduced along the text. The information in it must be critically presented in the text.

Figure 5- remove

Figure 4 – correct an error; membrane disruption?

Remove/change/correct sentences (some have unnecessary information for a review Ms.) like the ones (some examples):

“They are proteins which break down and convert complicated compounds to produce simple products. Enzymes are produced by different types of living cells ranged from eukaryotes to prokaryotes. Fungi are among living cells that produce enzymes.

Enzymes are ideal metabolites, which present in nature as macromolecular protein ingredients that provides the existence of several endogenous biochemical reactions via a well-defined process in the environment”.

“They play an important role in lowering the energy of activation and accelerate numerous biological reactions that are crucial to nourishing. (…) Enzymes are produced by all types of live cells including prokaryotes and eukaryotes. They are produced by plants, animals and microorganism such as fungi, bacteria and viruses.”

“The most important of enzymatic activity for the environment is produced extracellularly by fungi, owing to their ability to decay of inanimate materials??, whereby fungal enzymes play an important role in the process of lignin degradation by ligninase and laccase. This application makes fungi not consider as parasites in the environment but can be also considered as saprobes.”???

- Correct some spelling mistakes: rot (instead of rote); wild (instead of wilde)

Author Response

Comments and Suggestions for Authors

Your review approaches an important and an actual, theme: enzymes from fungi. Despite this, the manuscript (Ms.) must be deepen in some sections (3 and 4), other eliminated (section 2) or edited (section 1). As yeasts are completely absent from the text (just one or two notes in table 4), the title should reflect that.

Response

At first, we appreciate your reviewing feedbacks and your careful reading of our manuscript, and interest in our study. We appreciate the opportunity to clarify our research novelty. We have revised all the manuscript and have made all necessary changes according to your comments and recommendations.

We improved sections 3 and 4 as well as we edited section 1.

Section in our point of view is important because it give brief informations about structure and functions of fungal enzymes.

Point 1- change several sentences: remove or change some statements.

Response

The manuscript was revised and improved.

Point 3 – deepen the information – give examples, conditions, compare species, phyla, for instance; add chitinases.

Response

The manuscript was supported with recent and deeper information.

Also, chitinases were included in details.

Point 4 - deepen the various sub-sections; basidiomycetous fungi are completely absent from the text.

Response

This point was covered in the revised manuscript.

I did not understand the inclusion of 4.3.2 and 4.3.3 in the text. Of course, in point 4.3.2, enzymes are behind the synthesis of almost all substances – from that point of view all fungal enzymes are important.

Response

In sections 4.3.2. and 4.3.3., we gave a brief glance about the important applications of fungal enzymes in medical uses.

The information in Table 4 is not presented/explained, introduced along the text. The information in it must be critically presented in the text.

Response

Table 1 was cited in the text.

Figure 5- remove

Response

Thank you for your suggestion.

We change figure 5 to suitable for represent the mechanism of anticancer effect of the fungal enzymes.

Figure 4 – correct an error; membrane disruption?

Response

This error was corrected.

Remove/change/correct sentences (some have unnecessary information for a review Ms.) like the ones (some examples):

“They are proteins which break down and convert complicated compounds to produce simple products. Enzymes are produced by different types of living cells ranged from eukaryotes to prokaryotesFungi are among living cells that produce enzymes.

Enzymes are ideal metabolites, which present in nature as macromolecular protein ingredients that provides the existence of several endogenous biochemical reactions via a well-defined process in the environment”.

“They play an important role in lowering the energy of activation and accelerate numerous biological reactions that are crucial to nourishing. (…) Enzymes are produced by all types of live cells including prokaryotes and eukaryotes. They are produced by plants, animals and microorganism such as fungi, bacteria and viruses.”

“The most important of enzymatic activity for the environment is produced extracellularly by fungi, owing to their ability to decay of inanimate materials??, whereby fungal enzymes play an important role in the process of lignin degradation by ligninase and laccase. This application makes fungi not consider as parasites in the environment but can be also considered as saprobes.”???

Response

Thank you for your comments.

The manuscript was revised.

Correct some spelling mistakes: rot (instead of rote); wild (instead of wilde)

Response

The manuscript was revised.

Round 2

Reviewer 2 Report

Thanks for the changes, I think the quality of the paper has been improved.

There are just mistakes/typos that should be corrected. Species name should be always in italic. Moreover, the name of the enzymes sometimes has an initial capital letter; this is not necessary.

4.2.1. Bioremediation. Authors focused on just polyethylene and hydrocarbons. They can be used as an example, but the chapter should have a more general view. Among plastic polymers, PE is certainly a problem, but it is not the only problem. In the same regards, PHAs are an issue, but pesticides, EDCs, drugs as well. You can check the literature and eventually use a table to resume the major findings.

Line 543-544: mistakes to be corrected

Reviewer 3 Report

In general, the manuscript has been improved, most of the comments have been taken into account.
It would be prudent to separate the purpose in the introduction as a separate paragraph.
I think this review will be helpful.